# Unidirectional Finite Control Set-Predictive Torque Control of IPMSM Fed by Three-Level NPC Inverter with Simplified Voltage-Vector Lookup Table

**Ibrahim Mohd Alsofyani *** and **Laith M. Halabi**

Department of Electrical and Computer Engineering, Ajou University, World Cup-ro 206, Yeongtong-gu,
Suwon 16499, Republic of Korea
* Correspondence: alsofyani@ajou.ac.kr; Tel.: +82-10-9560-0049

**Abstract:** This paper proposes a unidirectional finite control set-predictive toque control (UFCS-PTC) method for a three-level neutral-point-clamped (3L-NPC) inverter fed interior permanent magnet synchronous motor (IPMSM). The proposed algorithm can lower the complexity of PTC fed by 3L-NPC by reducing the number of admissible voltage vectors (VVs) effectively. The candidate VVs are restricted within 60° of the voltage space voltage diagram (VSVD), which is the nearest to the flux trajectory for each 60° flux sector. After the segmentation of the VSVD and flux trajectory, the proposed method can keep VVs in one direction during the prediction process, which can result in significant torque/flux reduction. Therefore, the UFCS-PTC can reduce the number of admissible VVs from twenty-seven to six while achieving excellent steady-state performance in terms of reduced flux and torque ripples. Additionally, the proposed method eliminates the need for weighting factor calculation for neutral point voltage associated with a 3L-NPC inverter. The UFCS-PTC of IPMSM also has other features, such as improved balancing capability of the DC-link capacitors' voltage, small computation time due to the reduced number of admissible voltage vectors considered in the cost function, and easy implementation. The effectiveness of the proposed method is verified through experimental results.

**Keywords:** model predictive control; permanent magnet synchronous motor; three-level neutral-point clamped (3L-NPC) inverter

## 1. Introduction

Recently, interior permanent magnet synchronous machines (IPMSMs) have been widely applied for many industries and electrified transportation owing to their distinguished merits, such as high-power efficiency, robustness, and relatively simple torque control [1–3]. Therefore, it has attained rapid progress as a high-performance and highly efficient motor.

One of the most used control approaches for AC motor drives is field-oriented control (FOC), which uses the proportional-integral (PI) controllers for indirectly controlling the flux and torque over the direct and quadrature currents [4,5]. Despite its good quality, however, this method requires tedious tuning of the PI controller gains. Additionally, the dynamic response is limited by the bandwidth of the internal current-control loop [6].

Recently, finite control set model predictive control (FCS-MPC) has been applied for high-performance IPMSMs owing to its attractive features such as easy inclusion of non-linear constraints, simple implementation, intuitive concept, and fast dynamic response. In this method, the discrete model of the motor is used to predict the motor variables based on tracking the reference signals to obtain the optimal voltage vector (VV) using a cost function to achieve the optimal control performance [7–11]. FCS-MPC can be further categorized into FCS-predictive current control (FCS-PCC) and FCS- predictive torque control (FCS-PTC).

Compared to the two-level inverters, the three-level neutral-point-clamped (3L-NPC) inverter is a widely employed multilevel inverter in industries. It is advantageous for higher efficiency, less total harmonic distortion (THD), and less switching stress across switching devices [12,13]. Nevertheless, designing the FCS-MPC of an AC machine fed by a 3L-NPC inverter requires 27 different voltage vectors to be enumerated by the cost function for every control period resulting in high computation burden [14–17]. In addition, the 3L-NPC inverter suffers from the inherent neutral point (NP) voltage balance, which causes higher stress on some switching devices and large output waveform distortion [14]. Hence, FCS-MPC must take into account another extra constraint or optimization objective to balance the NP voltage of the 3L-NPC system.

To overcome the abovementioned problem, a long horizon prediction algorithm can be applied to enhance steady-state performance and lower the current distortions in MPC with a 3L-NPC inverter-fed drive. Nevertheless, the computational load of the optimization problem for a 3L-NPC system increases exponentially with the length of the prediction horizon [18,19]. Instead, a one-step horizon becomes preferable for prediction.

For lowering the control and computation complexity, there are several studies that can be categorized under lookup table [8,20] and deadbeat control [21–27] methods to reduce the computation load. The deadbeat control method of FCS-PTC is highly parameter-dependent and complex as it requires the derivation of voltage reference based on the flux and torque [23–25]. In deadbeat control, the optimum location of the space vector diagram sector is obtained by a reference voltage. However, the reference voltage is used to determine the sector or sub-sector of the space vector diagram where the voltage vectors in the selected sector or subsector need to be evaluated in the cost function. Therefore, even if the reference voltage vector is determined, it is still needed to define the closest VVs to be assessed by the cost function.

The performance of FCS-PTC using VV lookup tables is much simpler, and it varies according to structure and operating conditions. Ref. [26] proposed sequential lookup tables depending on the geometrical analysis of VVs located in the voltage space vector diagram (VSVD). However, this method needs various VV enumeration stages, and the control complexity rises with the increasing number of VVs. Refs. [27,28] proposed the use of flux and torque errors or solely the torque error to restrict the number of VV enumerations.

Another challenge for FCS-PTC with a 3L-NPC system is the addition of NP voltage objective in the cost function. As the number of objectives increases, the cost function design becomes more challenging as the number of weighting factors increases. Thus, considering this issue in the 3L-NPC system, control schemes have been proposed to auto-tune or eliminate weighting factors of the cost function [26,29–31]. Ref. [29] proposed a direct MPC approach for a 3L-NPC-fed motor drive with an online weighting factor adjustment using a fuzzy-logic strategy. In [26,30], the sequential FCS-PTC was introduced, respectively.

However, existing FCS- PTC methods with a 3L-NPC inverter still use a large control set of admissible VVs during the prediction stage and apply complicated algorithms to eliminate the weighting factors, thereby increasing computational load and control complexity. Additionally, the conventional methods still suffer from an increase in torque ripple and stator flux ripple.

Therefore, this paper proposes a new and simplified VV selection approach for improving the performance of FCS-PTC of 3L-NPC inverter-fed IPMSM. This proposed method (UFCS-PTC) focuses mainly on lowering the complexity of PTC by reducing the number of admissible VVs greatly and effectively. Hence, the number of enumerations of VVs is reduced from twenty-seven VVs to six per control cycle, which contributes to a significant reduction in computation time. Since the tuning for weighting factors is quite tedious, a simple balancing method for the capacitive voltages is applied to eliminate the need for a balancing objective in the cost function. The proposed method still maintains the advantages of the conventional method, such as the inclusion of non-linear constraints, simple implementation, and fast dynamic response. The effectiveness of the proposed method was validated experimentally.

This paper is organized as follows: Section 2 presents the fundamentals of a 3L-NPC system. The classical PTC method is expressed in Section 3. The principles of the proposed PTC are clearly explained in Section 4. Experimental results justify the feasibility of the proposed strategy in Section 5. Finally, some conclusions are made in Section 6.

**2. Fundamentals of 3L-NPC Inverter**

Figure 1 shows the configuration of the IPMSM-fed 3L-NPC inverter. Each leg of 3L-NPC inverter has four switching devices: $S_{x1}$, $S_{x2}$, $S_{x3}$, and $S_{x4}$ (x = a, b, and c). As can be depicted in the voltage space vector diagram (VSVD) of the 3L-NPC inverter shown in Figure 2, there are 27 various voltage vectors (VVs) with a grouping of 27 switching states (SSs). This includes three zero VVs (ZVVs) $V_0$, 12 small VVs (SVVs) $V_1, \ldots V_6$, 6 medium VVs (MVVs) $V_7, \ldots V_{12}$, and 6 large VVs (LVVs) $V_{13}, \ldots V_{18}$. Depending on the connections to the top capacitor voltage ($V_c^T$), bottom capacitor voltage ($V_c^B$), or neutral point (NP), ZVV has three SSs, SVV has two opposite SSs, and MVVs and LVVs with a single SS. *P*, *O*, and *N* indicate the phase connections to the '+' DC voltage bus, NP, and '−' DC bus, respectively. The relation between the phase connection and SS is shown in Table 1. The NP *O* is connected to each leg by clamped switches, which divide the dc-bus into the upper and lower parts with the voltage denoted by $V_c^T$ and $V_c^B$, respectively. Unbalanced DC-link capacitor voltages can result in high current THD and cause serious effects on control performance, such as increasing the torque and flux ripples. To ensure a balanced 3L-NPCI drive system, the voltages $V_c^T$ and $V_c^B$ should be evenly distributed. Thus, it is essential to determine the NP voltage $\Delta V_c$ for satisfactory system performance as,

$$\Delta V_c = V_c^T - V_c^B \tag{1}$$

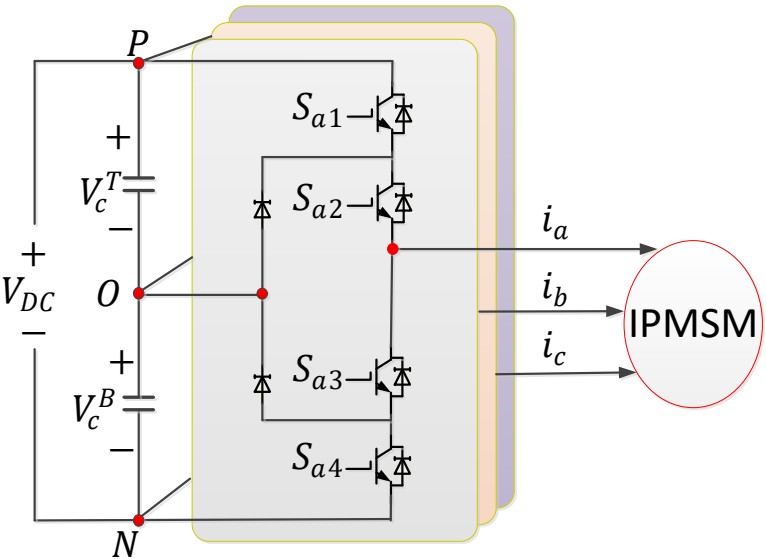

**Figure 1.** Configuration of the 3L-NPCI with IPMSM drive.

**Table 1.** Switching states of 3L-NPC Inverter.

| Switching State | Device Switching Status (x = a, b, and c) | | | | Pole Voltage |
|---|---|---|---|---|---|
| | $S_{x1}$ | $S_{x2}$ | $S_{x3}$ | $S_{x4}$ | |
| *P* | 1 | 1 | 0 | 0 | $V_{DC/2}$ |
| *O* | 0 | 1 | 1 | 0 | 0 |
| *N* | 0 | 0 | 1 | 1 | $-V_{DC/2}$ |

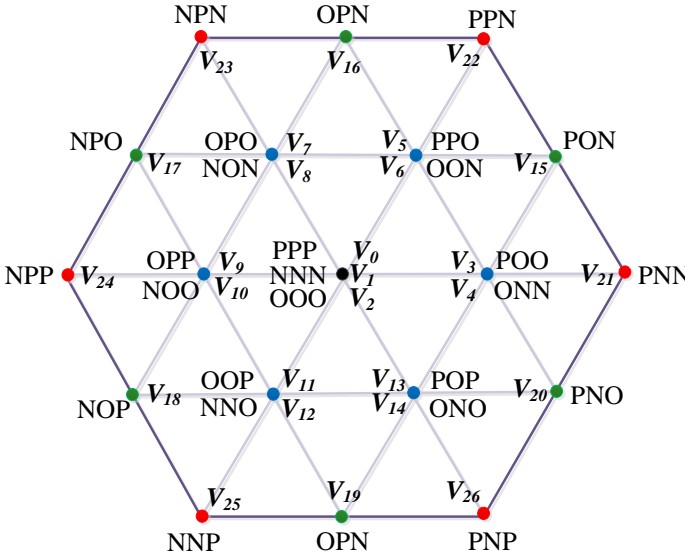

**Figure 2.** Voltage space-vector diagram of a three-level inverter.

The NP voltage $\Delta V_c$ can be predicted by using the present measured currents, measured capacitor voltages, and applied optimal switching state [14]. In most previous research studies, the NP voltage is incorporated as an objective in the cost function of PTC to solve the balancing problem of the NPC inverter.

## 3. Conventional PTC Algorithm for 3L-NPC Inverter

This method assumes that the motor is symmetrical and the motor parameters are constant. Hence, the mathematical model of IPMSM in the stationary reference frame is described using forward Euler's approach as [8]:

$$\vec{\lambda}_s(\mathrm{k}) = L_s \cdot \vec{i}_s(k) + \vec{\lambda}_{pm}(k), \tag{2}$$

$$T_e = \frac{3}{2}p \cdot \left( \vec{\lambda}_s(\mathrm{k}) \cdot \vec{i}_s(k) \right), \tag{3}$$

where $\vec{i}_s$ and $\vec{\lambda}_s$ denote the stator current and stator flux vectors. $\vec{\lambda}_{pm}$ denotes the permanent magnet flux. $T_e$ is the electrical motor torque. $R_s$ and $L_s$ are the stator resistance and stator self-inductance, respectively. $P$ is the number of pole pairs. $L_s$ is the synchronous inductance. In the IPMSM case, $d$- and $q$-axis stator inductances ($L_d$, $L_q$) are not equal to $L_s$ (i.e., $L_d \neq L_q$).

To predict the stator flux at the next control instant, the following equations are used as [8,20],

$$\vec{\lambda}_s(k+1) = \vec{\lambda}_s(\mathrm{k}) + T_s \cdot \vec{v}_s(k) + T_s R_s \cdot \vec{i}_s(k) + T_s \cdot Y \cdot \omega_r(k), \tag{4}$$

where $\vec{v}_s$ and $\omega_r$ represent the stator VV and motor rotor speed, respectively. $\vec{i}_s = \begin{bmatrix} i_{s\alpha} \\ i_{s\beta} \end{bmatrix}$, $\vec{v}_s = \begin{bmatrix} v_{s\alpha} \\ v_{s\beta} \end{bmatrix}$, and $Y = \begin{bmatrix} \lambda_{s\beta} \\ -\lambda_{s\alpha} \end{bmatrix}$.

It is also essential to predict the stator current. Hence, the current prediction is expressed as [8]:

$$\begin{bmatrix} i_{s\alpha}(k+1) \\ i_{s\beta}(k+1) \end{bmatrix} = \begin{bmatrix} 1 - R_s T_s/L_d & L_q \cdot \omega_r T_s/L_d \\ -L_d \cdot \omega_r T_s/L_q & 1 - R_s T_s/L_q \end{bmatrix} \begin{bmatrix} i_{s\alpha}(k) \\ i_{s\beta}(k) \end{bmatrix} + \begin{bmatrix} T_s/L_d & 0 \\ 0 & T_s/L_q \end{bmatrix} \begin{bmatrix} v_{s\alpha} \\ v_{s\beta} \end{bmatrix} + \begin{bmatrix} T_s/L_d & 0 \\ 0 & T_s/L_q \end{bmatrix} \begin{bmatrix} v_{s\alpha} \\ v_{s\beta} \end{bmatrix} + \begin{bmatrix} 0 \\ \lambda_{pm}(k) \cdot \omega_r T_s/L_q \end{bmatrix}, \tag{5}$$

where $T_s$ denotes the sampling time, $k$ and $k + 1$ denote the present and predicted control discrete cycles, respectively.

Based on the predicted flux and current, the electric torque can be obtained as below:

$$T_e(k + 1) = \frac{3}{2} p \cdot \left( \vec{\lambda}_s(k + 1) \cdot \vec{i}_s(k + 1) \right). \tag{6}$$

As previously mentioned, the 3L-NPC structure consists of 27 VVs. In a 3L-NPC inverter, it is necessary to balance NP voltage besides minimizing the flux and torque errors. Therefore, to determine the optimal VV among the admissible VVs, a cost function that includes three objectives is formed [14,22]:

$$g = \left| T_{ref} - T_e(k + 1) \right| + W_\lambda \left| \lambda_{ref} - | \lambda_s(k + 1) | \right| + W_{cv} \lfloor \Delta V_c \rfloor, \tag{7}$$

where $T_{ref}$ is a torque reference, and $\lambda_{ref}$ is the flux reference at the $k^{th}$ sampling time. $W_\lambda$ and $W_{cv}$ denotes the weighting factors that identify the relative values of the flux control and the NP voltage.

Thus, after enumerating the 27 VVs, the voltage vector, which can minimize the cost function $g$, is the optimal voltage vector $V_{opt}$.

## 4. Proposed PTC Algorithm

The structure of the UFCS-PTC is shown in Figure 3. The proposed method still uses the same stages of estimation, prediction, and cost function evaluation as in the classical PTC. To obtain the best performance of 3L-VSI fed IPMSM, the optimization priority for the proposed PTC algorithm must be focused on the tracking of flux and torque references trajectory, while the neutral point potential needs to be balanced using the redundant switching states after the optimal VV is obtained. Hence, unlike previous literature, the proposed PTC can reduce the complexity of the control algorithm by reducing the number of weighting factors, as the NP voltage objective is not involved in the cost function.

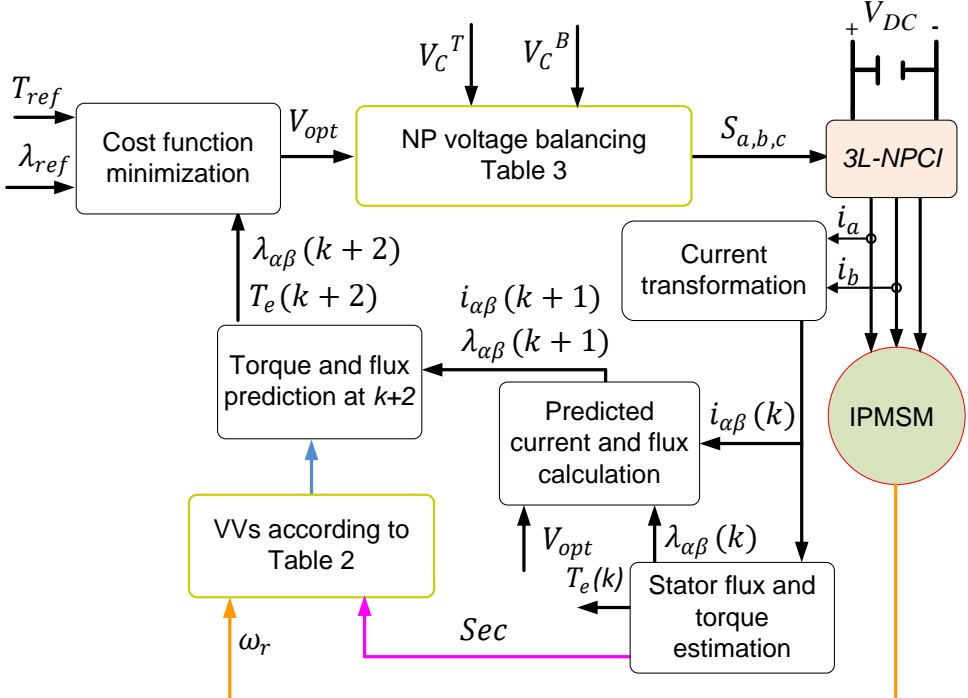

**Figure 3.** UFCS-PTC of PMSM fed by 3L-NPC inverter.

### 4.1. Proposed Preselection Method

The main purpose of the proposed PTC method is to improve the performance of the FCS-PTC method for a 3L-NPC inverter system while reducing the number of admissible VVs, thus, reducing the computation load on the digital signal processor (DSP). The flux trajectory is evenly distributed into six sectors.

As the position of the stator flux, $\phi_s$, is estimated as

$$\phi_s = \arctan\left(\lambda_\beta / \lambda_\alpha\right). \tag{8}$$

Therefore, each sector in the $\alpha - \beta$ plane of flux rotation is determined according to the following equation

$$(2N - 3)\cdot\pi/6 \leq Sec(N) \leq (2N - 1)\cdot\pi/6, \tag{9}$$

where $Sec$ is the flux sector and $N = 1, \ldots, 6$. The optimum directions of active VVs that ensure the circular motion of stator flux according to the flux position should be determined.

By neglecting the stator resistance, the change in the stator flux vector $\Delta\psi_s$ can be expressed as [8]:

$$\Delta\lambda_s = \vec{v}_s\, T_s. \tag{10}$$

Hence, it is obvious that the variation in stator flux $\Delta\lambda_s$ is controlled by the VV and hence, is proportional to the direction of the applied VV during $T_s$. similar to the segmentation of the flux trajectory, the VSVD for the 3L-NPC inverter is also evenly divided into six symmetrical segments (i.e., $R_1$–$R_6$), as shown in Figure 4. In the proposed VSVD, there are 19 various VVs with a grouping of 27 SSs.

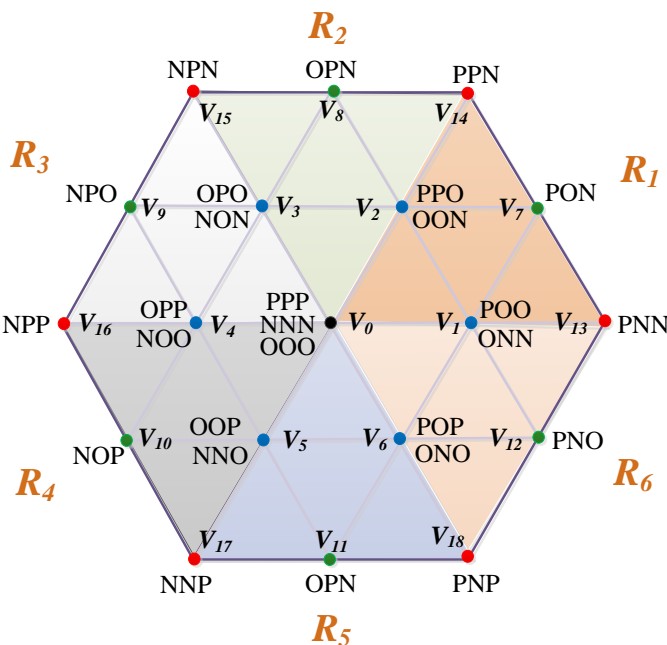

**Figure 4.** Proposed voltage space voltage vector diagram.

In this way, each flux sector will correspond to one VSVD region, which contains the nearest VVs to the circular flux trajectory, as shown in Figure 5. Notably, the proposed method emphasizes the reduced enumeration of admissible VVs to minimize the computation burden on the DSP.

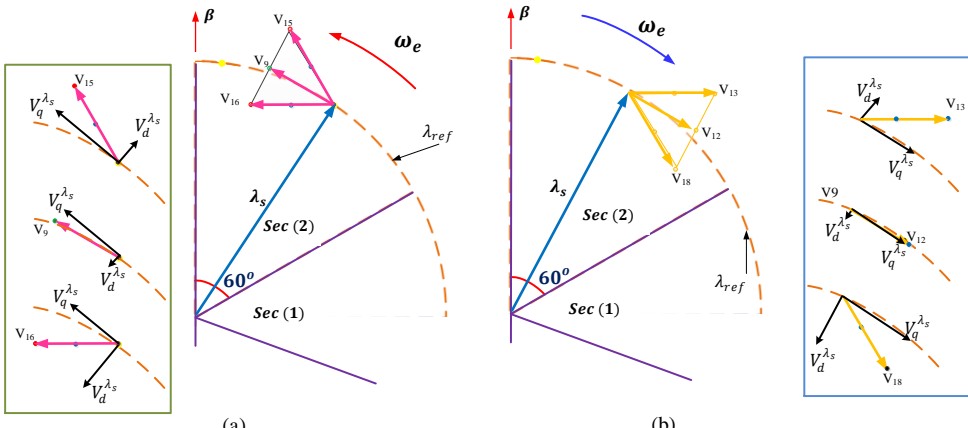

(a)                                        (b)

**Figure 5.** Illustration for the optimal candidate VVs for Sec (2) at (**a**) anticlockwise speed direction and (**b**) clockwise speed direction.

Figure 5 shows the moving stator flux $\lambda_s$ at the instantaneous location of Sec (2) during the anticlockwise (forward) and clockwise (reverse) motor speed directions. It is apparent from Figure 5a,b that the optimal VSVD regions containing the admissible VVs are different for both speed directions. $R_3$ is the optimum region in the forward direction, and the $R_6$ is the optimal VSVD region in the reverse direction. For convenience, Table 2 shows the optimum VSVD regions for each flux sector. It can be observed that only six VVs out of nineteen VVs are processed during the cost function evaluation in each speed direction. The normal routine for controlling torque and flux is to include flux and torque error conditions for the preselection of VVs [26,28]. Nevertheless, this work uses the motor speed $\omega_r$ for both VVs prediction selections because the rotor speed and flux rotation travel in the same direction [8]. The speed can lead to lower control complexity and simultaneously control both stator flux and electrical torque. Hence, the predicted VVs will be restricted in one direction, and the optimal control for torque and flux can be satisfied using the admissible VVs in the selected VSVD region.

**Table 2.** Optimal VVs for each flux sector.

| Flux Sector | Optimal VVs | |
| --- | --- | --- |
| | **Forward Motor Direction** | **Reverse Motor Direction** |
| *Sec* (1) | $[V_0, V_2, V_3, V_8, V_{14}, \text{ and } V_{15}]$ | $[V_0, V_5, V_6, V_{11}, V_{17}, \text{ and } V_{18}]$ |
| *Sec* (2) | $[V_0, V_3, V_4, V_9, V_{15}, \text{ and } V_{16}]$ | $[V_0, V_1, V_6, V_{12}, V_{13}, \text{ and } V_{18}]$ |
| *Sec* (3) | $[V_0, V_4, V_5, V_{10}, V_{16}, \text{ and } V_{17}]$ | $[V_0, V_1, V_2, V_7, V_{13}, \text{ and } V_{14}]$ |
| *Sec* (4) | $[V_0, V_5, V_6, V_{11}, V_{17}, \text{ and } V_{18}]$ | $[V_0, V_2, V_3, V_8, V_{14}, \text{ and } V_{15}]$ |
| *Sec* (5) | $[V_0, V_1, V_6, V_{12}, V_{13}, \text{ and } V_{18}]$ | $[V_0, V_3, V_4, V_9, V_{15}, \text{ and } V_{16}]$ |
| *Sec* (6) | $[V_0, V_1, V_2, V_7, V_{13}, \text{ and } V_{14}]$ | $[V_0, V_4, V_5, V_{10}, V_{16}, \text{ and } V_{17}]$ |

The example presented in Figure 5 can explain further the working principle of the method. When the flux linkage $\lambda_s$ moving anticlockwise at $Sec(2)$ according to Table 2, the candidate VVs in VSVD region $R_3$ $[V_0, V_3, V_4, V_9, V_{15}, \text{ and } V_{16}]$ are used for controlling the flux and torque. For the analysis, we can use the radial VV component $(v_d^{\psi_s})$, which is perpendicular to the flux trajectory $(\lambda_{ref})$ and tangential VV component $(v_q^{\psi_s})$, which is tangential to the flux trajectory. Notably, the radial voltage component is accountable for the rise and reduction in flux, whereas the tangential voltage component is responsible for the increase and decrease in torque. It is apparent from Figure 5a that the tangential components for three candidates VVs ($V_9$, $V_{15}$, and $V_{16}$) in segment 3 ($R_3$) are pointing in the anticlockwise direction, indicating the increase in torque. Whereas the small VVs ($V_3$, $V_4$) and zero VV ($V_0$) can be applied when reducing torque. At the same time, it can be observed that the radial component of $V_{15}$ is greater than the flux trajectory, while the

radial component of $V_{16}$ is smaller than the flux trajectory. This indicates the increase or decrease in flux demand can still be satisfied with the same admissible VVs.

Similarly, in Figure 5b in the same flux location but for the opposite speed direction, the VSVD region 6 ($R_6$) with the candidate VVs [$V_0$, $V_1$, $V_6$, $V_{12}$, $V_{13}$, and $V_{18}$] is selected to satisfy the flux and torque demands in the reverse direction as can be shown from the radial and tangential components of $V_{12}$, $V_{13}$, and $V_{18}$. The same scenario can be applied to all flux-VSVD regions in Table 2.

### 4.2. Neutral Point Voltage Balancing

It is well known that unbalanced dc-link capacitor voltages can result in high current distortions and causes serious effects on control performance, such as increasing the torque and flux ripples. Hence, the NP voltage should be maintained within an acceptable fluctuation range of around zero volts.

In this paper, to ensure balanced dc-link capacitor voltages without a need for additional weighting factors, the opposite SSs of the SVVs are proposed as in Table 3. The switching between SSs will depend on the capacitance-voltage magnitude. There is only one weighting factor for the flux control objective. In this way, the control complexity owing to the tedious tuning of multiple weighting factors is reduced. Hence, the cost function in the proposed method can be defined without considering the NP voltage objective as

$$g = \left| T_{ref} - T_e(k+1) \right| + W_\lambda \left| \lambda_{ref} - |\lambda_s(k+1)| \right|. \tag{11}$$

**Table 3.** Optimal SSs of small VVs for capacitance-voltage balancing.

|  | $V_{opt}$ ($V_1$) | $V_{opt}$ ($V_2$) | $V_{opt}$ ($V_3$) | $V_{opt}$ ($V_4$) | $V_{opt}$ ($V_5$) | $V_{opt}$ ($V_6$) |
|---|---|---|---|---|---|---|
| $V_c^T \geq V_c^B$ | [POO] | [PPO] | [OPO] | [OPP] | [OOP] | [POP] |
| $V_c^T < V_c^B$ | [ONN] | [OON] | [NON] | [NOO] | [NNO] | [ONO] |

### 4.3. Proposed PTC Algorithm

The overall control sequence for the proposed PTC is summarized as follows:

① Estimate the stator flux, $\vec{\lambda}_s(k)$ and electrical torque, $T_e(k)$.
② Obtain the proposed admissible VVs based on the speed direction depending on flux position according to Table 2.
③ Predict the stator current, $\vec{i}_s(k+1)$, stator flux, $\vec{\lambda}_s(k+1)$, and electrical torque, $T_e(k+1)$ by applying the optimal VV, $V_{opt}$.
④ Then, evaluate the predicted variables by calculating the cost function using (11).
⑤ Select the optimal VV, $V_{opt}$ which results in a minimum cost function.
⑥ Balance neutral point voltage using opposite SSs of the SVVs as in Table 3.
⑦ Apply the optimal VV with the suitable SS at the next sampling time.

## 5. Experimental Results

The performance of the proposed PTC (UFCS-PTC) for the 3L-NPC inverter-fed IPMSM coupled with an induction machine (load) was implemented experimentally using a functional DSP (TMS320C28335). The experimental setup is shown in Figure 6. For capturing results, a Lecroy oscilloscope (HDO6104, 1 GHz, 2.5 GS/s) was used. The specifications of the machine parameters are provided in Table 4. The reference flux and weighting factor were set at 0.27 Wb and 150 for both PTC methods. The sampling time ($T_s$) is 100 μs. The DC link voltage was supplied by 300 V.

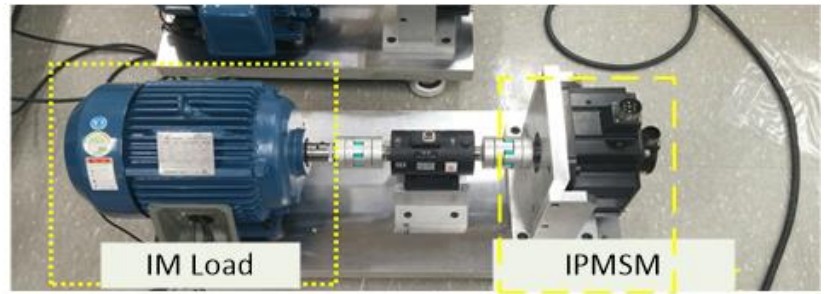

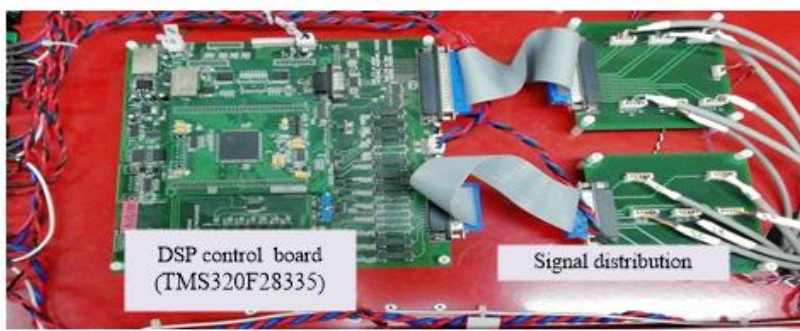

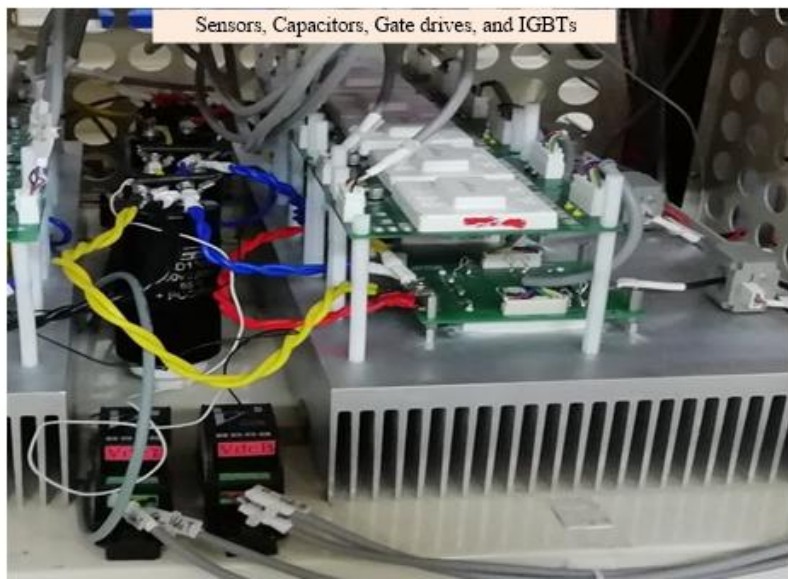

**Figure 6.** Experimental setup.

**Table 4.** PMSM Parameters.

| Parameter | Value |
|---|---|
| Torque rating | 27 Nm |
| Power rating | 5.5 kW |
| Current rating | 11 A |
| Speed rating | 1750 rpm |
| Permanent magnet flux | 0.264 Wb |
| Number of poles | 8 |
| Stator resistance | 0.158 Ω |
| *d*-axis self-inductance | 7.29 mH |
| *q*-axis self-inductance | 7.25 mH |

The proposed method is compared with the multistep PTC (MSPTC) method in [26] for a 3L-NPC inverter with 19 VVs. To have a fair comparison, the MSPTC will use the same proposed NP voltage balancing method and the same sampling time. Hence, both methods will manipulate the selection for the 19 VVs, whereas the redundant switching states for the small VVs will depend on the proposed capacitors' voltages balancing method after the optimal VV is selected.

Figure 7 shows the steady-state performance at 100 rpm with a load of 5 Nm for the UFCS-PTC and MSPTC [20]. From top to bottom, the curves shown in Figure 7 are flux magnitude, electrical torque, capacitor voltage error $V_{c.err}$, line-to-line voltage $V_{ab}$, and A-phase output current $i_a$. The torque and flux ripples are calculated online using the equation in (12) [32].

$$X_{ripple} = \sqrt{\frac{1}{n}\sum_{i=1}^{n} X(i) - X_{av}}, \ X_{av} = \frac{1}{n}\sum_{i=1}^{n} X(i) \tag{12}$$

where $n$ is the number of samples, and $X$ refers to torque and flux. $X_{av}$ denotes the average quantity.

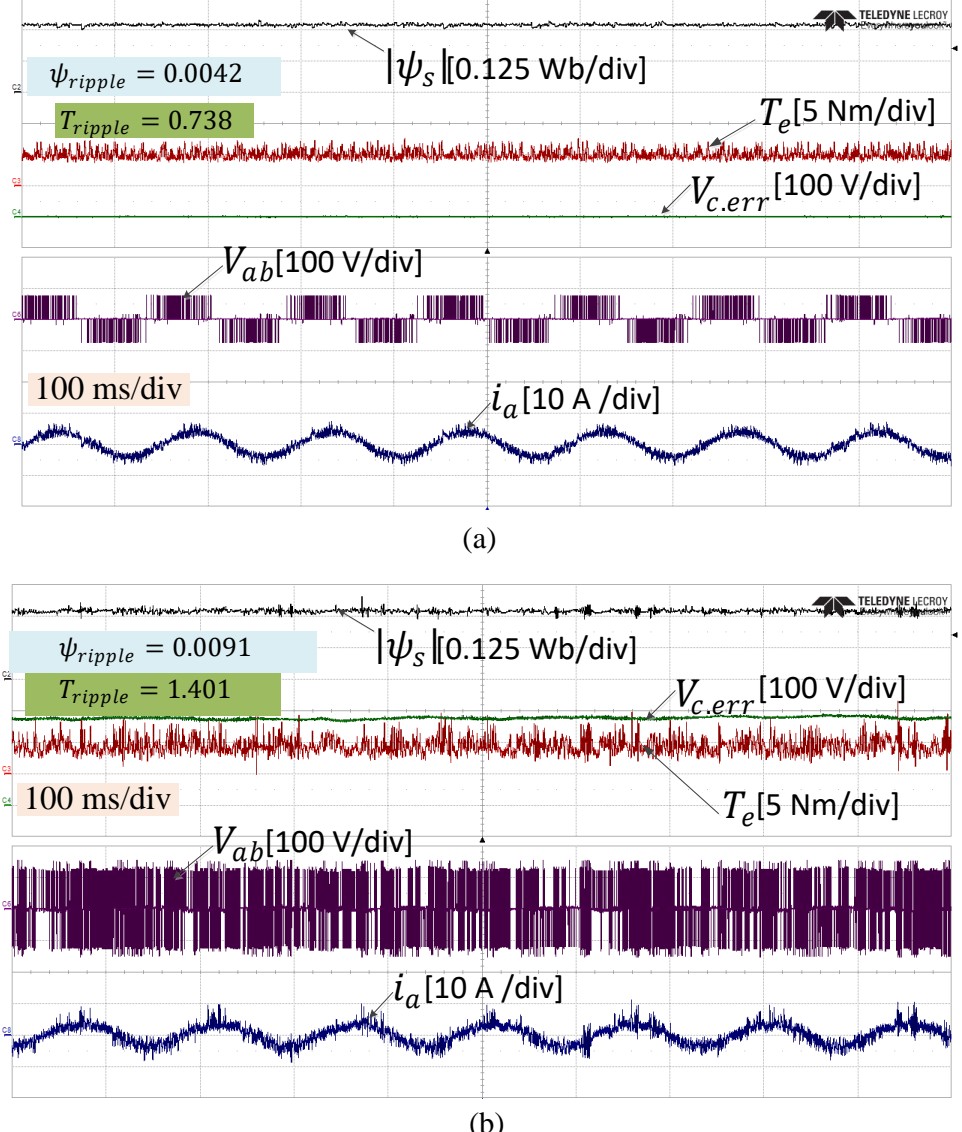

(a)

(b)

**Figure 7.** Experimental steady-state response at 100 rpm for (**a**) UFCS-PTC and (**b**) MMPTC.

It can be seen from the results that the proposed PTC has significantly improved steady-state performance with less torque and flux ripples while obtaining very small capacitor voltages error, which indicates no voltage deviation between the top and bottom dc-link capacitor voltages. Furthermore, the line-to-line voltage $V_{ab}$ of the proposed method has a better voltage waveform shape with reduced voltage magnitude when compared to the MSPTC as a result of the small VV selection. It can be observed from Figure 7b that the capacitor voltage error is very large, resulting in unbalanced capacitor voltages, which indicates that the MSPTC has less selection for the small VVs responsible for the balance of capacitance voltages compared to the proposed PTC. It is worth mentioning that the balancing control is applied for both PTC methods. The large increase in the capacitor error contributed to the large torque and current ripples for the MSPTC method.

Figure 8 shows the steady-state performance of both UFCS-PTC and MSPTC methods at 600 rpm with a load of 10 Nm. Once again, the proposed method shows a smaller flux and torque ripple in relation to the MSPTC method. Compared to the MSPTC, the proposed method has also reduced voltage spikes for the line-to-line voltage. It can be easily seen that the proposed UFCS-PTC has almost zero capacitor voltage error, whereas the MSPTC still suffers from deviation of capacitance voltages.

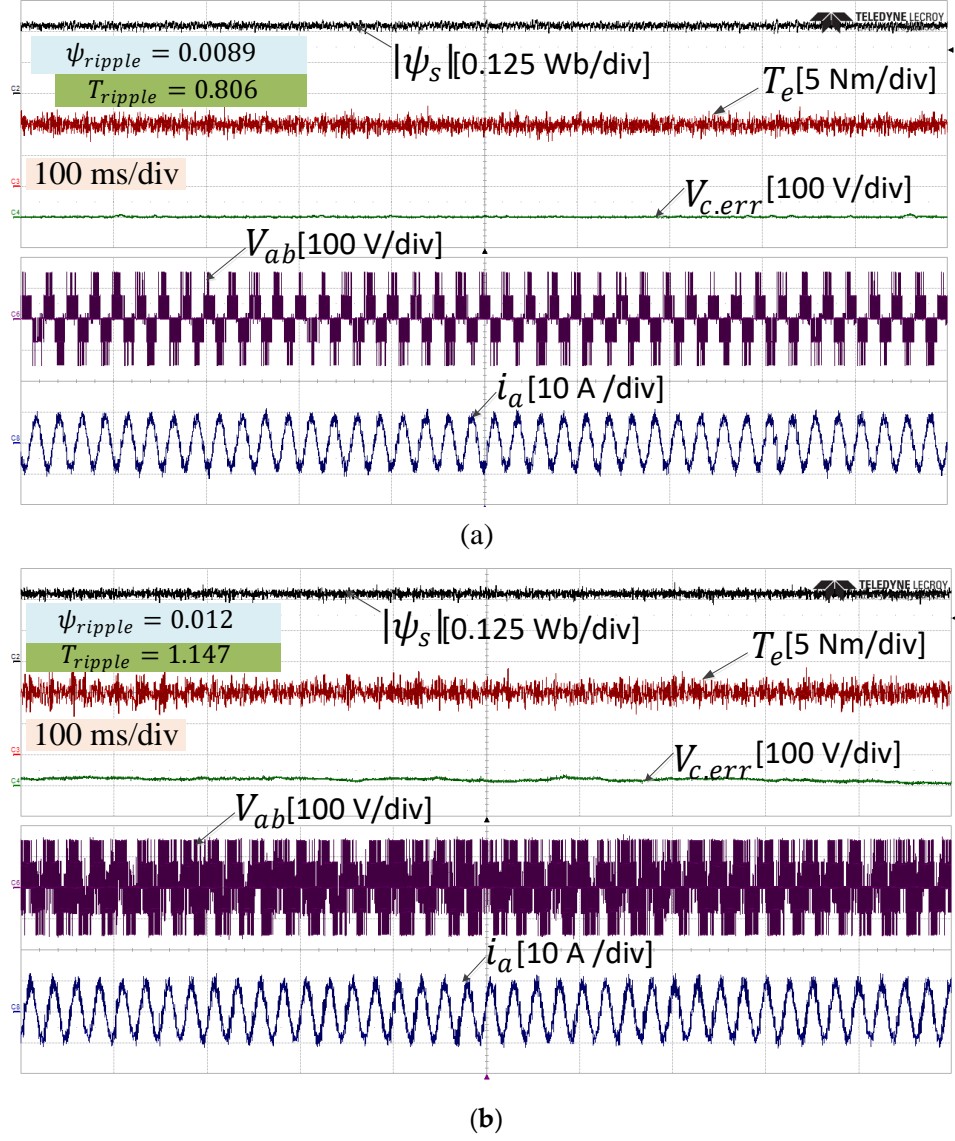

**Figure 8.** Experimental steady-state response at 600 rpm for (**a**) UFCS-PTC and (**b**) MMPTC.

Figures 9 and 10 show the spectral waveforms for both UFCS-PTC and MSPTC schemes for both speed and torque operations in Figures 7 and 8. It is apparent that the proposed method has fewer harmonics than MSPTC for low-speed operation. At high-speed operation, it can be observed that the harmonics for UFCS-PTC are condensed with some spikes within the first 1 kHz. Nevertheless, the conventional method contains larger harmonic components throughout the spectrum. Thus, the proposed method has a smaller THD than that of MSPTC.

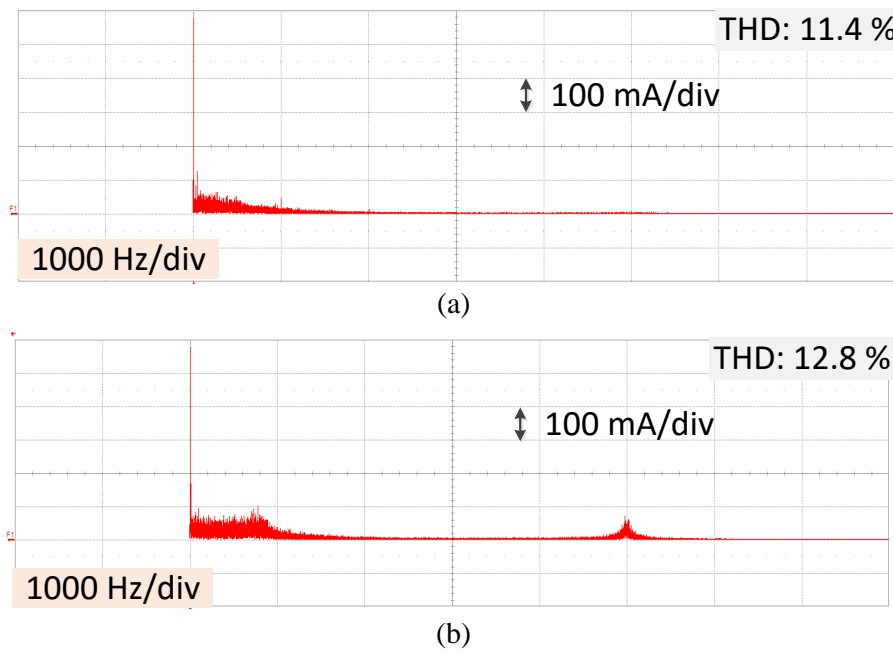

**Figure 9.** Experimental frequency spectra of A-phase motor current at 100 rpm. (**a**) MSPTC, (**b**) UFCS-PTC.

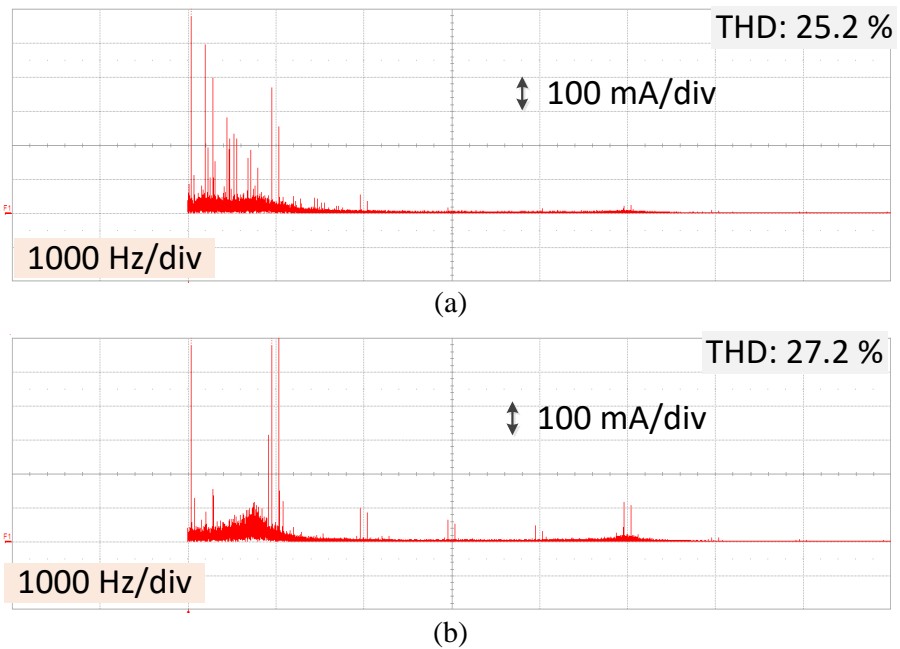

**Figure 10.** Experimental frequency spectra of A-phase motor current at 600 rpm. (**a**) MSPTC, (**b**) UFCS-PTC.

Figure 11 shows the computation burdens for UFCS-PTC and MSPTC methods, respectively. They were obtained by the input/output bins of the DSP controller. It is evidently seen that the proposed PTC method can significantly reduce the computation burden to 19.8 μs, whereas the computation of MSPTC is about 33.2 μs because the preselection strategy for the UFCS-PTC method can limit the admissible VVs from twenty-seven to six enumerations per control cycle.

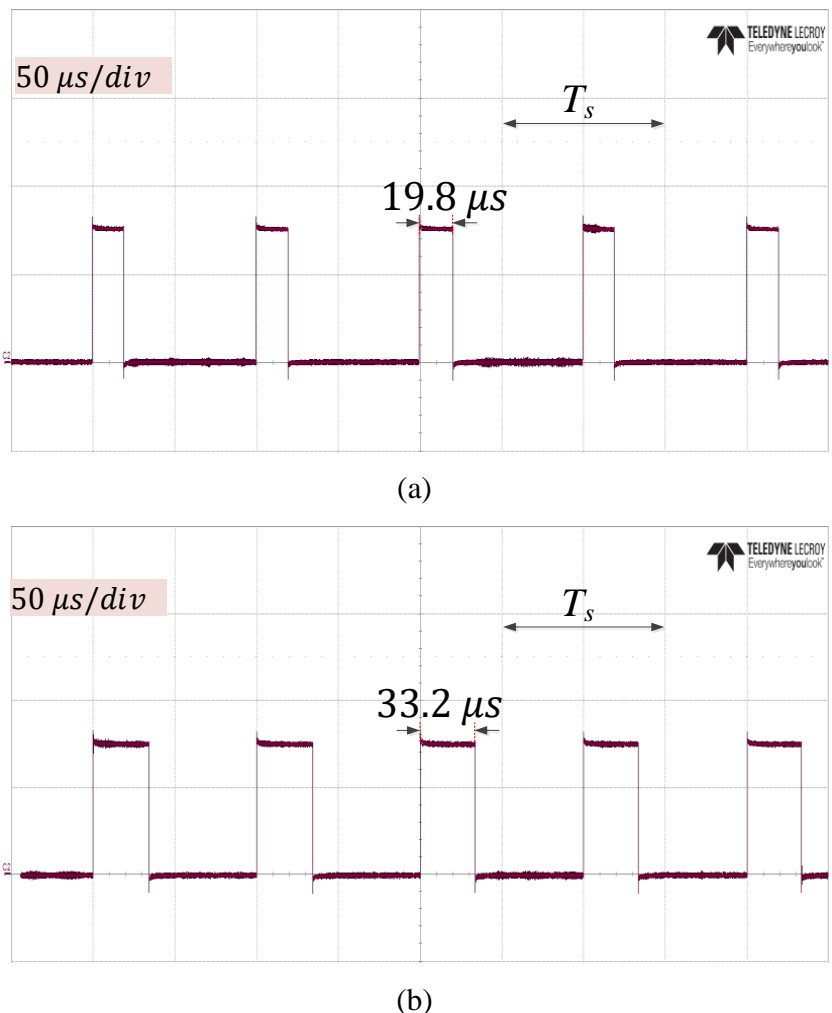

(a)

(b)

**Figure 11.** Computation burden. (**a**) UFCS-PTC, (**b**) MSPTC.

Finally, Figure 12 shows the torque-transient performance from 2 to 10 Nm for both UFCS-PTC and MSPTC methods when the IPMSM runs at 600 rpm. Apparently, both methods demonstrated equal and excellent torque transient performance with a transient time of 0.92 ms. Nevertheless, it is clear that the toque ripple of MSPTC increases by more than twice when the reference torque changes to 10 Nm (see Figure 12b). Nevertheless, the proposed PTC method maintains an approximately similar torque ripple in both torque references, as shown in Figure 12a.

For the result summary, the quantitative comparison between UFCS-PTC and MSPTC in terms of torque and flux ripples, THD, number of admissible VVs, and computation time is presented in Table 5. Apparently, the proposed method has superior performance in all of these aspects. In addition, it has well-regulated NP voltage for all operating conditions. This confirms the effectiveness of the proposed method and its suitability for industrial applications.

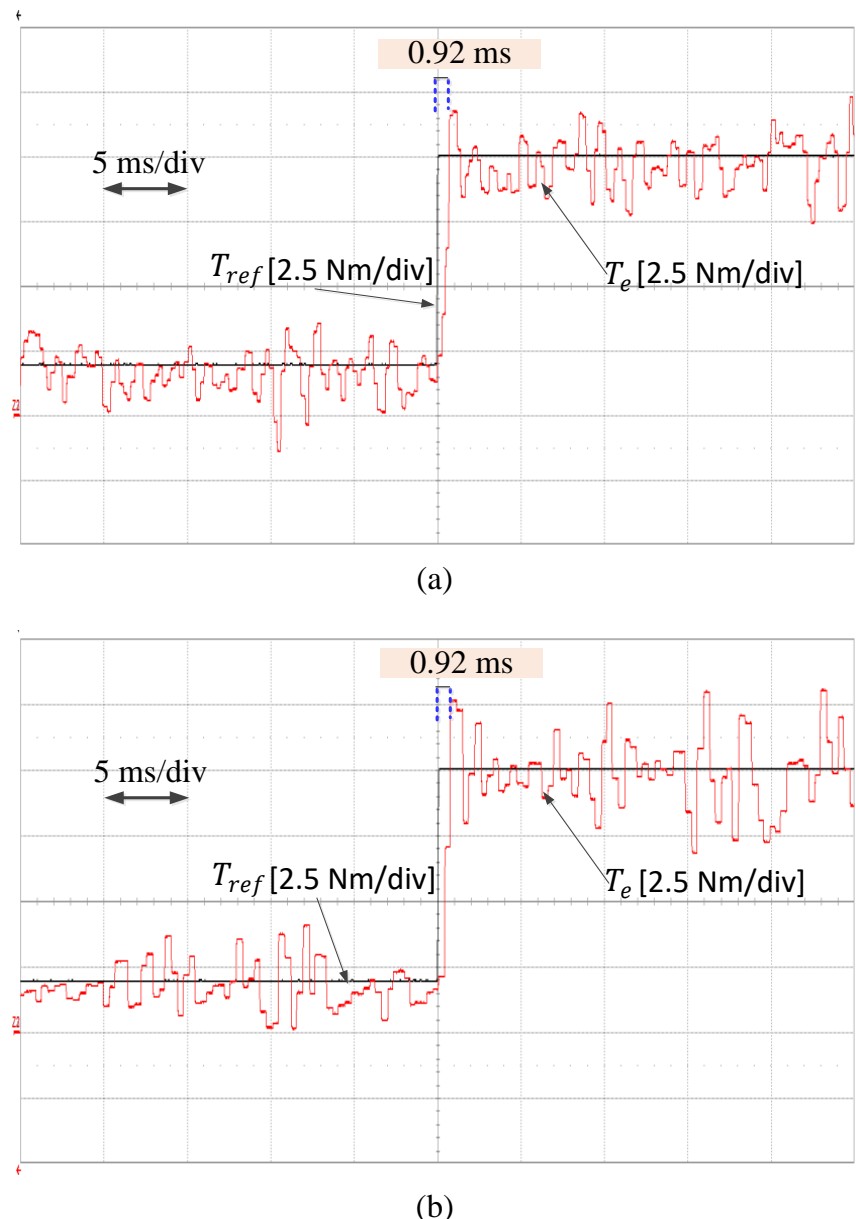

**Figure 12.** Experimental dynamic torque response from 2 to 10 Nm at 600 rpm for (**a**) UFCS-PTC and (**b**) MMPTC.

**Table 5.** Quantitative comparison for UFCS-PTC and MSPTC.

| Operating Condition | | Method | Torque Ripple (Nm) | Flux Ripple (Wb) | THD (%) | No. of Admissible VVs | Comp. Time (μs) |
|---|---|---|---|---|---|---|---|
| Speed (RPM) | Torque (Nm) | | | | | | |
| 100 | 5 | MSPTC | 1.401 | 0.0091 | 12.8 | 13 | 33.2 |
| | | UFCS-PTC | 0.738 | 0.0042 | 11.4 | 6 | 19.8 |
| 600 | 10 | MSPTC | 1.147 | 0.012 | 27.2 | 13 | 33.2 |
| | | UFCS-PTC | 0.806 | 0.0089 | 25.2 | 6 | 19.8 |

## 6. Conclusions

This paper proposed a unidirectional finite control set-predictive torque control method to improve the IPMSM drive fed by the 3L-NPC system. The proposed method was simply designed to reduce the number of VVs from twenty-seven to six in the 3L-NPC inverter using speed and flux information with a simple lookup table. Additionally, the

problem of neutral point voltage in the 3L-NPC inverter system was simply resolved without using a weighting factor. A comprehensive analysis of the need for unidirectional VVs in optimizing the performance of the proposed PTC was presented. The results of the proposed scheme depict excellent steady-state performance compared to the conventional PTC. Additionally, it has comparable dynamic torque performance to that of the conventional PTC method. Compared to the conventional PTC, the proposed method managed to reduce the computation time by approximately 40%. The effectiveness of the proposed technique was verified through the experimental results.

**Author Contributions:** Author Conceptualization and writing—original draft, formal analysis, resources I.M.A.; review and editing and validation, I.M.A. and L.M.H. All authors have read and agreed to the published version of the manuscript.

**Funding:** This research received no external funding.

**Acknowledgments:** Special thanks go to Kyo-Beum Lee for permission to use the Power Electronics Equipment.

**Conflicts of Interest:** The authors declare no conflict of interest.

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
