# Peer review of "Unidirectional Finite Control Set-Predictive Torque Control of IPMSM Fed by Three-Level NPC Inverter with Simplified Voltage-Vector Lookup Table"

_electronics, doi:10.3390/electronics12010252_

Round 1
Reviewer 1 Report
I still found the following problems:
1. I strongly advise rewriting the abstract to more accurately capture the general theme of this article. Perhaps, points such as results and discussion, and conclusion should be included in this abstract.
2. Kindly, please define the abbreviations (such as VVs, PMSM, etc) before using them. Although the author did define some abbreviations under the Abstract, they should be defined again under the Introduction. Treat the abstract as a separate manuscript.
3. How is predicted from Equation (4)? In Equation (4) are stator current in the discrete case?
4. What meaning of K?
5. Where does the cost function of the optimal VV (7) come from? Is it proposed by yourself? More explanations are needed.
6. The Proposed MPTC Algorithm in the paper introduce is new or already exist? How can authors check whether the cost function conditions are satisfied in practice? Suppose your proposed algorithm is more efficient, that is mean It possible provides to compare it with other MPTC algorithms.
7. The experimental part is mainly about the simulation experiment of the proposed method, which lacks the comparison results with other methods. It is suggested to increase the experimental content.
8. The language used throughout this paper needs to be improved, the author should do some proofreading on it. Give the article a mild language revision to get rid of a few colloquial sentences and complete the sentences.
Author Response
The authors appreciate your valuable comments and expertise. The paper quality is really improved based on your constructive comments.

Reviewer 2 Report
This paper proposes a unidirectional finite control set-predictive toque control (UFCS-10 PTC) method for a three-level neutral-point-clamped (3L-NPC) inverter fed permanent magnet synchronous motor. The main concerns are listed as follows.
(1) The proposed control methods should be described more clearly. What is the key innovation compared with conventional method?
(2) Please provide more details on the key parameters selection of the experimental case.
(3) Experimental results are given to prove the feasibility of the proposed control strategy. More failure scenarios are encouraged to verify the effectiveness.
(4) At section “Conclusion”, the authors are recommended to denote in detail the summary and prospects, and the wider applicability perspectives of their research outcomes.
Author Response

(The authors gave the same response as above.)

Reviewer 3 Report
The paper titled “Unidirectional Finite Control Set-Predictive Torque Control of PMSM fed by Three-Level Neutral-Point-Clamped Inverter” has proposed a unidirectional finite control set-predictive toque control (UFCS PTC) method for a three-level neutral-point-clamped (3L-NPC) inverter fed permanent magnet synchronous motor. The proposed algorithm can lower the complexity of PTC by reducing the number of admissible VVs effectively.
I have following comments for the paper: -
1. The introduction section has to be modified as there are many references cited in a single line. For Example: “For lowering the control and computation complexity, there are several studies which can be categorized under deadbeat control and lookup table methods to reduce the computation load [14]-[22].” The literature in the reference has to be analyzed critically.
These 2 latest reference can also be helpful in the study, please analyze these 2 critically:
(i) ‘Reactive Power Compensation for Grid by Packed-U-Cell Inverter Using Model Predictive Control Strategy with Intelligent Multi-objective Scheme’. 1 Jan. 2022 : 793 – 806.
This paper proposes a model predictive control strategy for 15 level Packed-U-Cell inverter that satisfies multiple-objectives of low current total harmonic distortion (THD), capacitor voltage balances, supply of desired active and reactive power, as well as lower switching and lower voltage stresses on the switching devices.
(ii) “Model predictive control of Packed U-Cell inverter for microgrid applications,” Energy Reports, Elsevier, https://doi.org/10.1016/j.egyr.2022.05.188
In this paper, the PI voltage and current controllers have also been modelled on a seven-level PUC inverter. Thereafter, model predictive control of five, seven and fifteen level inverters are also formulated and then simulated using MATLAB/Simulink. Finally, HIL validation of different PUC inverters is performed.
2. At the end of the introduction section, structure of the paper is required.
3. Authors have reported that “the proposed MPTC can reduce the complexity of control with reducing the number of weighting factors as the NP voltage objective is not involved in the cost function.” Can these numbers be quantified ..? If yes, then please explain by how much %age the complexity is reduced, by how many numbers the weighing factors is reduced.
4. A comparison of results in the form of Table should be presented in the results section.
Author Response

(The authors gave the same response as above.)

Reviewer 4 Report
1) You wrote: “Recently, permanent magnet synchronous machines (PMSMs) have been widely researched and applied for many industries owing to its distinguished merits, such as high-power efficiency, robustness and relatively simply torque control [1].”, but there is no conclusion.
2) Why did you decide to implement finite control set model predictive control? From your explanation it just because “becomes broadly applied”. You’d better focus on the advantages over conventional techniques.
3) What was the requirement of your project? Why does the conventional techniques are not applicable?
4) What is “MPTC”? Denote acronyms before usage.
5) When you develop or provide mathematical model, you always have to focus on assumptions, which were made. Otherwise, the model is void.
6) You discuss PMSMs in general, however they have different Ld and Lq, while you consider them equal.
7) In the description of experimental setup, measuring equipment should be fully specified.
8) There is a poor explanation regarding experiment plan and results. Please explain, which experiments did you perform and why. After that provide numerical results for each experiment. You stated ripple reduction of XX%, however measured ripples are not specified.
9) How does your algorithm handle non-linearities, say saturation?
10) What is the efficiency of motor drive under proposed control comparing to the conventional PI control?
11) Did you involve (or plan to involve) any of MTPA techniques to increase system efficiency in higher load region, where saturation impacts motor inductances?
The list of method can be found in:
A. Dianov, F. Tinazzi, S. Calligaro and S. Bolognani, "Review and Classification of MTPA Control Algorithms for Synchronous Motors," in IEEE Transactions on Power Electronics, vol. 37, no. 4, pp. 3990-4007, April 2022, doi: 10.1109/TPEL.2021.3123062.
Author Response

(The authors gave the same response as above.)

Round 2
Reviewer 3 Report
All comments are addressed.
Author Response

(The authors gave the same response as above.)

Reviewer 4 Report
1) When you respond to reviewer’s comments, you’d better provide a modified text as well. Otherwise it is hard to detect the modification. Furthermore, some of comments were not replied properly, or reported paper modifications are not found.
2) Some of your figures are drawn in raster format. Please use vector format, especially for schematics
3) My previous comment “3) What was the requirement of your project? Why does the conventional techniques are not applicable?” was not properly replied. You have to explain specific of your project and why you decided to develop the proposed technique, why the existing ones are not applicable? Ex. We develop motor drive for low-cost applications, therefore computation intensive algorithms are not applicable.
4) I asked you to denote acronyms before usage, however it is still not fixed! The same acronym is used without explanation: “MPTC”
5) You did not properly answer to my comment: “5) When you develop or provide mathematical model, you always have to focus on assumptions, which were made. Otherwise, the model is void.” You did not explain conditions, at which your model equations are correct, which is not technically sound approach.
Ex. Will you model work in saturation? How does it consider eddy currents? Is it true to non-symmetrical machines?
6) Comment “6) You discuss PMSMs in general, however they have different Ld and Lq, while you consider them equal.” is not properly answered. PMSMs by definition are Permanent Magnet Synchronous Machines, which generally have different Ld and Lq. Thus, you have to note it! The machines with equal inductances are subclass of PMSM, say, surface mounted PMSM. Please pay more attention to the technical statements.
7) The comment “8) There is a poor explanation regarding experiment plan and results. Please explain, which experiments did you perform and why. After that provide numerical results for each experiment. You stated ripple reduction of XX%, however measured ripples are not specified.” Is not replied properly.
The answer about method: “formula from [22]” is not technically SOUND!!! What is the number of formula? Is it applicable for your conditions? Furthermore, I di not find any formula for torque ripples there.
8) If you do not consider non-linearities in your work, you have clearly state it in conditions!
However, if not, it seems that you do not use the main advantage of your method. Over conventional PI control.
9) When you develop new method, you typically improve several parameters and worsen others. Therefore, you have to evaluate improvements and worsening. For this reason I asked about efficiency comparing to the conventional PI control
Author Response

(The authors gave the same response as above.)

Round 3
Reviewer 4 Report
Dear authors,
I am really dissatisfied with your approach to responding my comments and paper modifications. You do not respond to my comment properly and sometimes ignore them, especially in the experimental part. I hope you can pay more attention to the improvement of your manuscript.
1) You mentioned PMSM with almost similar Ld and Lq, which is incorrect, because in general case PMSMs have different Ld and Lq. I asked you to check this issue and find out the type of motor, which seems to be surface mounted PMSM, where Ld and Lq are typically equal (or similar). However, you wrote that the motor was interior type PMSM, which usually have significantly different Ld and Lq. How can you explain it?
From Fig.6 it can be clearly seen that the test motor is a commercial type, thus its model name MUST be provided. Actually, I requested to specify the equipment in the previous round of review, but you did not do it for motor.
2) Equation (12) is incorrect!!! From this equation ripples (Xripple) are ALWAYS zero. Furthermore, the units are incorrect: the left part is in X units (say Nm), while the right part is in square root of X units (say root(Nm))!!!
3) I complained that only some pictures are in vector format, while it is necessary for all schematics and diagrams. Instead of improving raster pictures, you worsen all the pictures converting them into tiff, being rater format.
4) You stated that one of the most important advantages of the PTC-based methods is ability to control machines with NON-LINEARITIES, however you consider LINEAR motor in mathematical modelling, which is not technically sound.
5) The comment 9 left unreplied!! I asked for the numerical evaluation of the developed method to understand its advantages and disadvantages. You HAVE to evaluate experimentally IMPROVEMENTS and WORSENINGS of the proposed technique and compare to the existing techniques. Mentioning general tendencies in the introduction is not a technically SOUND approach.
Author Response
Thanks for the efforts and time for improving the quality of the paper.
